# Effectiveness, Challenges, and Environmental Impacts of New Food Strategies with Plant and Animal Protein Products

**DOI:** 10.3390/foods13203217

**Published:** 2024-10-10

**Authors:** Ziane da Conceição das Mercês, Natalia Maldaner Salvadori, Sabrina Melo Evangelista, Tatiana Barbieri Cochlar, Virgílio José Strasburg, Vanuska Lima da Silva, Viviani Ruffo de Oliveira

**Affiliations:** 1Post Graduate Program in Food, Nutrition and Health, Federal University of Rio Grande do Sul (UFRGS), Porto Alegre 90035-003, RS, Brazil; zianemerces@gmail.com (Z.d.C.d.M.); natisalvadori18@gmail.com (N.M.S.); sabrina.evangelista@ufrgs.br (S.M.E.); tatianabarbieri2010@hotmail.com (T.B.C.); 2Department of Nutrition, Federal University of Rio Grande do Sul (UFRGS), Porto Alegre 90035-003, RS, Brazil; vstrasburg@hcpa.edu.br (V.J.S.); vanuska.lima@ufrgs.br (V.L.d.S.)

**Keywords:** sustainable agriculture, meat substitutes, analogous to meat, vegetable protein, animal protein

## Abstract

Sustainable food practices are intrinsically linked to human nutrition in the preservation of the ecosystem. This study, therefore, evaluates the effectiveness, challenges, environmental impacts, and new food strategies related to plant and animal products, with a view to promoting more sustainable and healthy eating practices. The search stages were conducted using the following databases: PubMed, Science Direct, and SciElo. The studies selected included those published from 2018 to 2024 and government documents, available in English, Portuguese, and Spanish. The 34 articles analyzed in this study showed the environmental impacts related to the production of plant and animal proteins, highlighting the urgency of implementing changes in this sector. However, factors such as land use, carbon footprint, and water footprint show remarkable differences depending on the type of crop cultivated, agricultural practices adopted, and stages involved in the supply chain. As final considerations, the analysis suggests that achieving sustainability in food systems requires an integrate approach that combines the optimization of plant protein production with a reduction in environmental impacts and the development of technologies that that support the efficiency and resilience of the industry. Meeting the nutritional needs of the population in a sustainable way will only be possible through regional actions and a deep understanding of the challenges and opportunities.

## 1. Introduction

Sustainable food practices in the current scenario are intrinsically linked to human nutrition, seeking to reconcile the preservation of ecosystems with an adequate supply of macronutrients. Among these, protein supply stands out as one of the biggest challenges. In other words, achieving global food security and implementing sustainable food systems has proven to be an increasing challenge in the field of international policies. Given its relevance, the global food system emerges as a key element in several Sustainable Development Goals (SDGs) [1,2].

According to Elkington and Rowlands [3], sustainability is based on three dimensions: social, economic, and environmental. These aspects form the basis of the triple bottom line concept, which aims to satisfy the resource needs of present and future generations while ensuring the protection of the environment [4].

In addition, the concept of food sustainability is also highlighted through the Planetary Health Diet as part of the EAT-Lancet Commission and presents itself as an expressive opportunity to review dietary patterns [5]. The choices people make regarding their food not only affect their health and well-being, but also the future of the planet.

The livestock sector, for example, is described as one of the main causes of climate change, presenting itself as a burden on the environment [6,7,8,9,10]. Additionally, it is estimated that the meat industry will expand its production by 50–73% to meet the global population, which will reach 9 billion people by 2050 [9]. Based on these points, questions arise about the excessive use of natural resources, ethical concerns related to animal welfare, and the possible impacts on human health and the environment, which have become the central concerns regarding the consumption of animal-based proteins [8].

In addition, new strategies have emerged as substitutes for traditional animal meat, such as analogs to plant-based meat, edible insects, cultivated meat, 3D-printed meat, modified meat, which are gaining more and more attention as possible solutions for minimizing these impacts, making it possible to change the traditional models of production and consumption of food from animal sources [7]. The development of plant-based meat analogs is an example of a simple strategy that could mitigate the problems mentioned above. Since 2014, the business model around these proteins has been classified as a global food trend [8,9,11].

Whatever the way to reduce excessive meat consumption is, it is being pointed to as an important step contributing to minimizing environmental impacts [12]. However, the impacts caused by plant production also need attention, as they also use soil, water, fertilizer products, energy, etc., which can impact ecosystems [13]. For Fogel [14], the use of pesticides and deforestation for the expansion of areas destined for cultivation and used both in the production of plant-based food and as animal feed are critical factors that impact the environment and the sustainability of food systems.

Therefore, this study aims to analyze the effectiveness, challenges, environmental impacts, and new food strategies related to plant and animal proteins.

## 2. Materials and Methods

A literature review on food sustainability was carried out with a focus on the efficiency of vegetable protein production in relation to animal protein. The search stages were conducted through a survey of the PubMed, Science Direct, SciElo databases, (Figure 1). The following keywords were used, specifically written in English: Vegetable protein; Animal protein; Meat plant-based; Meat analogues; Meat alternative; Food sustainability; 3D Proteins; Cultured meat protein; Insect proteins; Carbon footprint.

The inclusion criteria were as follows: (1) new strategies and approaches, including technological advances, sustainable agricultural practices, and public policy initiatives; (2) original research articles, systematic reviews, meta-analyses, and review articles; (3) publications in peer-reviewed journals; (4) innovative dietary strategies, such as plant-based processed foods.

The following exclusion criteria were defined: (1) patents, citations, letters to the editor, conference abstracts, and case reports; (2) studies involving animals as part of the research; (3) reports, theses, dissertations, and other documents that do not have scientific evidence; (4) websites, editorials, blogs, and other non-scientific sources; (5) duplicate studies.

Strategies and combinations were applied to search each scientific data platform with keywords that directed to the articles, including PubMed, Science Direct, and SciElo. The selected studies included scientific papers and government documents available in English, Portuguese, and Spanish. All references were organized and managed using the Mendeley software (version 2.116.1), and duplicate articles were eliminated. The titles, abstracts, and complete studies were reviewed to check for their relevance in relation to the research topic, using inclusion criteria aligned with the leading question.

## 3. Sustainability and Protein Products

The understanding of sustainability applied to the food industry is described as generally including concepts such as social justice, animal welfare, fair labor and trade, local agriculture, organic food production, and the concept of “natural”, just to mention the most prominent ones [15]. There is a major global challenge in guaranteeing food and nutrition security while preserving natural resources, such as land and water, due to climate change and population growth, in conjunction with changes in eating habits. Therefore, there is growing interest in sustainable and diversified food systems which can address these challenges more effectively [16].

Protein products are at the heart of this sustainable food system. In recent years, there has been an increase in demand for protein-related ingredients [17]. According to estimates, the global protein market was valued at around USD 38 billion in 2019, and projections indicate that this market is expected to grow at an annual rate of 9.1% in the period between 2020 and 2027 [16,17].

Among the main sources of protein are meats of animal origin. However, its excessive consumption has been indicated as a factor of great environmental impact in relation to an increase in lifestyle-related diseases. In the study by Alexander et al. [18], they highlight the need to modify dietary patterns, with an emphasis on reducing the consumption of beef and processed meat, to report these challenges.

Studies have shown that the food supply chain is responsible for a substantial portion of greenhouse gas (GHG) emissions, in addition to using an enormous quantity of natural resources such as water and land [19,20]. For example, for beef from beef cattle herds, 25% of the highest-impact producers account for 56% of the herd’s greenhouse gas emissions and 61% of land use (estimated at 1.3 billion metric tons of CO_2_-eq and 950 million hectares of land, mostly pasture).

The European Green Deal has highlighted the importance of a circular economy that respects planetary boundaries, while the Farm to Fork strategy aims to promote a healthier, safer, and more sustainable food system, from production to consumption, across the European Union [21]. An example of this pact is in the Danish dietary guidelines that have recently been updated to consider the climate impact of food systems, an approach also adopted by several other European countries, such as Italy, France, Germany, the Netherlands, and the United Kingdom [22,23].

In Asian a study by Feng et al. [24], a systematic meta-analysis on carbon footprint (CF) was conducted, and they observed that rice and beef have the highest carbon footprint among the major food categories, with CF average 1.31 kg CO_2_ eq kg^−1^ and 6.15 kg CO_2_ eq kg^−1^ (live weight), respectively. The CF represents the carbon gas emissions in CO_2_-eq per unit mass of product produced from various food groups, being quantified and compared on a global and continental scale [25].

In many parts of the world, decision makers are promoting a reduction in annual food consumption due to climate concerns. The suggestion is to encourage consumers to follow a more plant-based diet to reduce carbon emissions; nevertheless, plant-based diets as a strategy to minimize climate change have shown signs of declining, which may suggest a low sensory acceptance, high product cost, and the need to develop new plant-based protein options with improvements in sensory attributes [26,27].

In the face of growing concerns about food sustainability, climate change, and the demand for protein sources, several innovations are emerging in the field of food production. Edible insects have also been investigated as a nutritionally promising source that can help meet the growing demand for protein in a sustainable manner [28]. Regarding proteins from insects, the study by Smetana [29] carried out in Germany states that insects have recently become a viable alternative to food proteins in Western countries, being in full progression due to the high conversion rate of protein foods and potentially low environmental impact [28] and low impact on land use [29]. In the Swedish study by Berggren [30], the authors questioned the insect industry on how the food would be considered ecologically correct, and they drew attention to the empirical measure of ecological impact and sustainability between different insect production systems.

Another important fact regarding insect-based foods that is little-mentioned in the literature are the cases of food allergies; in the study by de Gier and Kitty Verhoeckx [31], the authors compiled data that demonstrate numerous cases of allergies resulting from the ingestion of foods derived from insects. According to the authors, from the case reports, it can be concluded that the patients who developed anaphylaxis after ingesting insects did not necessarily have an allergic history, which was the case in 54.3% (25/46) of the patients [31]. A Korean study by Ramachandraiah [32] presents data on black soldier fly farming, which is described as a promising food alternative; according to the authors, this production caused direct GHG emissions of about 17 ± 8.6 g of CO_2_ eq per kg of dry larval gain.

A Dutch study by Van Der Weele et al. [33] states that cultivated meat faces challenges in terms of sustainability due to high levels of processing. The authors describe that the same limitations are seen in highly processed plant-based meat alternatives, algae-based foods, and insects.

Three-dimensional food printing is an innovative technology with the potential to create food structures from a variety of materials. Recently, 3D printed meat analogs using alternative proteins such as plant and insect proteins have been developed. Despite the advances in taste and texture in this technology, they still cannot completely replicate the look and taste of a real steak [34].

For Van der Weele et al. [33], meat analogs are being studied and analyzed for their technological feasibility and production costs and sustainability [6]. According to Smith et al. [35], a disadvantage of plant-based protein sources is the presence of lower levels of essential amino acids compared to animal sources. The authors point out that while plant-based agriculture is considered to be an alternative to livestock, it may not be sustainable in terms of land use. Plant-based foods such as tofu and textured soy protein have been mainstays of Western eating for decades. In recent years, however, the highlight has been on the accelerated growth of plant-based meat analogs [13,36,37].

The adequacy of plant sources for supplying protein in nutrition, in terms of environmental sustainability and human health, depends on several factors. These include geographical location and climate, the cost of harvesting, transportation, the availability of processing techniques, and the extent of processing required [38]. For example, increasing the consumption of vegetable protein is essential to promote a more sustainable use of natural resources, considering that, on average, the production of 1 kg of animal protein requires 5 kg of vegetable protein [39].

According to Parajuli et al. [40], the contributions to GHG emissions throughout the life cycle of the different stages of the fruit, cereal, and legume supply chain are diverse. For instance, in a tomato supply chain, the carbon footprint related to the cultivation phase ranged between 31% and 58%, while the packaging subsystem contributed between 7% and 54%, depending on the volume of fresh tomatoes. Parajuli et al. [40] used the same example of tomatoes to demonstrate the impact of tomato cultivation on the water footprint, about 94–98% of the impact (i.e., 21–30 L of water per kg of tomatoes). This impact was related to the production phase, and the rest was covered by the development of the nursery and the transportation of the product.

For Willett [5], approximately 30% of GHG emissions are emitted from food production, as well as 70% from freshwater use and the occupation of 37% of the earth’s ice-free surface. The authors further state that fundamental changes in the food system are needed to preserve environmental health. While collaborative action along the food chain is crucial to achieving the food system’s transformation, a detailed assessment of the potential contributions of each link in the chain is also needed.

Another idea commonly associated with food sustainability, but which is not always correctly understood by the public, is the concept of “food miles”. In this concept, people believe that transportation is the environmental impact factor of food products [15]. On food miles, the Australian study by Li et al. [41] analyzed the carbon footprint of food miles using a global multi-regional accounting approach. The research reveals that the transport sector is responsible for approximately 19% of the total emissions of the food system, which encompasses transport, production, and changes in land use. The global transport of goods related to the consumption of cereals and pulses contributes 36% of food mile emissions, almost double the amount of GHG emitted during their production [41].

As noted in those described above, food systems face unprecedented challenges in meeting the needs of a growing population, exacerbated by climate change [41]. The increasing concentration of CO_2_ in the atmosphere captures the infrared radiation that the Earth emits after absorbing sunlight, resulting in the warming of the planet. This process is behind the worrying trend in continuous rising global temperatures, both on the land surface and in the oceans, a phenomenon known as global warming [42,43].

For a better understanding of the carbon footprints left by the production of protein sources, Table 1 presents the CO_2_ emission values, comparing the products and the impacts caused to the environment.

The results presented in Table 1 reveal a discrepancy among the carbon footprints of different protein sources, highlighting the variation in the environmental impacts associated with each type of product. High temperatures can affect post-harvest quality in fruit and vegetable crops.

According to Leisner [56], high temperatures can favor photosynthesis in some crops, but they also have a negative impact on several aspects of post-harvest quality. They report that high temperatures can alter the flavor of fruits, influencing the sugar content in apples and grapes, as well as the acidity level of grapes. In addition, the firmness of fruits such as avocado and the oil content present in them can also be compromised under conditions of excessive heat.

In the study by Béné [57], the authors proposed a compilation of ideas in which they mention when food systems meet sustainability, along with current narratives and implications for action; according to the authors, the current food system is failing in several aspects crucial to global security and sustainability. It fails to predict the needs of the future world population, provide an adequate diet, ensure equitable benefits, and is contributing to environmental degradation. These failures threaten food security, nutritional health, social justice, and the sustainable use of natural resources.

Priority actions to correct these shortcomings include closing agricultural productivity and nutrient gaps, decentralizing food production with a focus on popular autonomy, and reducing the environmental impact of the food system by promoting greater efficiency in the use of energy, water, and carbon [57]. Nascimento and Tabai [58] reiterate that some measures can contribute to the reduction in environmental impacts, including reducing food losses and waste, consuming sustainably caught fish, and reducing the consumption of ultra-processed foods and sugary drinks.

### Effectiveness and Impacts of Vegetable and Animal Protein Production

The fundamental role of protein in life and daily life, in relation to food sustainability, is often associated with climate change, but relevant environmental issues go beyond this aspect. For Aiking and de Boer [59], several critical environmental limits have been established to avoid irreversible damage to the planet, such as the loss of biodiversity and the interruption of the nitrogen and carbon cycles. Food production, especially proteins, plays a central role in this scenario, contributing significantly to environmental degradation. The acceleration of the nitrogen cycle is one of the main factors affecting these processes, highlighting the interconnectedness between food production and broader environmental impacts [59].

Technological innovations in agriculture, from the Green Revolution to modern breeding techniques to precision and digital agriculture, have considerably increased crop yields in recent decades. However, despite these advances, the environmental and social costs have been high [60].

The environmental impact categories analyzed in the Serbia study by Skunca et al. [61] included Global Warming Potential (GWP), Ozone Depletion (DCO), Energy Demand (DE), Eutrophication Potential (PE), Acidification Potential (PA), and Land Use (US). For the complete system of extraction and isolation of RuBisCo protein from sugar beet leaves, the results were as follows: GWP of 16.41 kg CO₂-eq., DCO of 1.21 mg CFC-11-eq., DE of 205.24 MJ, PE of 4.73 g PO₄ P-lim, PA of 620.76 g SO₂-eq., and US of 0.19 m² of the organic agricultural area.

Another interesting point about environmental factors, such as temperature, aeration, humidity, and other factors, involved the fact that they exert a considerable influence on GHG emissions throughout the composting process, widely used in the preparation of land for planting and organic fertilizer [62,63].

According to Tisocco et al. [63], in 2020, agricultural GHG emissions were responsible for 11.4% of the total amount, of which 14.9% were related to the management of leachate, the mixture of feces, and urine produced and collected on cattle farms. An Irish study by Kavanagh et al. [64] reports that GHG production by agricultural manure management contributes up to 9% to global warming. Table 2 shows some studies that presented data on CO_2_ emission in the production of organic fertilizer.

Table 2 shows the variability in CO₂ emissions associated with different organic wastes used as fertilizers, including cattle, chicken, pig manure, and sewage sludge. These emissions vary according to the type of waste, the geographical location, and the measurement methods used. The results emphasize the importance of assessing the environmental impact of organic waste in specific contexts, highlighting the need for tailored approaches to managing and mitigating greenhouse gas (GHG) emissions.

Agricultural and industrial production, as well as domestic supply, consumes 7.709 km^3^ of fresh water globally, 87% of which is green water and 13% is blue water [70]. Agriculture is the main responsible for this consumption, accounting for 99% of global water use. Irrigation, used to compensate for soil moisture deficits, has contributed to the increase in agricultural productivity. However, this greater demand for water for irrigation has put pressure on water resources. With projected population growth and the growing preference for a diet rich in animal proteins, a increase from 60% to 110% in the production of food, feed, and biofuels will be required [70,71].

The Water Footprint (WF) of food products of animal origin is, in general, much higher than that of plant products. On average, beef has a WF 20 times higher than that of cereals and starchy roots. In addition, when considering the protein unit, the WF of beef is six times higher than that of legumes [72]. Healthy diets, however, do not always reduce WF from consumption, especially if animal products are replaced by foods such as fruits and legumes with relatively large WFs [71].

In a Brazilian study by Guimarães et al. [71] that analyzed the environmental footprints of food production, their findings were related to university menus, and the authors mentioned the use of the calculator “The Value of Food”, which calculates the values of food and the footprints, water, terrestrial (LF), and carbon (CF), left in the environment. According to the authors, the results showed that animal foods on menus were associated with higher levels of WF and CF, while vegetarian menus had lower levels of WF and CF.

Mekonnen and Hoekstra [70] observed that in the data on the water footprint of some food products, beef has the highest water consumption, requiring 15,415 L to produce 1 kg of product, standing out as the most demanding in terms of water resource use. In comparison, pork and chicken require much smaller volumes, with 5988 and 4325 L/kg, respectively. Among products of plant origin, legumes such as lentils (5874 L/kg) and dried beans (5053 L/kg) still have a relatively high consumption, although they are considerably less demanding than meat. Grains such as wheat (1827 L/kg), rice (1673 L/kg), and corn (1222 L/kg) require less water, standing out as more sustainable options from a water point of view. Overall, the data reinforce the idea that the production of plant-based foods, especially grains, uses much less water per kg of product, suggesting that diets with a greater emphasis on vegetables can relieve pressure on global water resources.

However, in the study by Mekonnen and Gerbens-Leenes [73], in a detailed analysis on the unsustainable use of unsustainable blue water, which represents the use of water from surface and groundwater sources in a non-renewable manner. In their results, the authors presented data in which wheat and rice stand out as the crops with the highest volumes of total blue WF, with 203,764 and 202,090 million m^3^/year, respectively. Wheat, meanwhile, has the largest unsustainable fraction, with 68% of its blue WF being unsustainable. Other important crops, such as cotton (67% unsustainable) and sugarcane (59%), also show a high impact on unsustainable water use, with contributions of between 10% and 8.5%, respectively. Forage crops, corn, and soybean follow the same trend, with fractions of unsustainable blue WF ranging between 53% and 65%.

## 4. Final Considerations

Knowing the efficiency, challenges, environmental impacts, and new dietary strategies related to plant and animal proteins was the objective of the analysis carried out throughout this study. It was observed that while plant proteins are a promising alternative to animal proteins, particularly in terms of sustainability, they still face challenges.

When it comes to environmental impacts, it was found that, compared to the production of animal proteins, those of plant origin use fewer natural resources. Land use, carbon exchange, and hydrological rates vary depending on the type of crop, the agricultural practices used, and the stages of the supply chain. This study also shows that greenhouse gas emissions are present at all stages of production, from agriculture to transportation, with direct implications for global warming.

Nutritional specifications, such as the lower levels of essential amino acids found in plant sources, are central aspects, as well as technological predictions of meat analogs and production costs.

New food practices, such as composting and technological advances in agriculture, offer a means of mitigating the environmental impacts related to food production. To ensure their effectiveness, however, it is imperative that these strategies consider local conditions and environmental and social variables.

Therefore, the analysis suggests that achieving sustainability in food systems requires an integrated approach that combines optimizing plant protein production with reducing environmental impacts and developing technologies that support industry efficiency and resilience. Meeting the nutritional needs of the global population in a sustainable way will only be possible through regional action and a deep understanding of the challenges and opportunities.

The analysis of global freshwater consumption reveals that agriculture is mainly responsible for this use, accounting for 99% of the total, with a great dependence on irrigation in order to increase productivity. However, this practice has generated significant pressure on water resources, especially in the face of predicted population growth and growing demand for animal proteins, which will require a substantial increase in food production. The Water Footprint of animal products is notably higher compared to vegetables, with beef requiring 15,415 L per kilogram, while pulses and grains have considerably lower values, standing out as more sustainable options. In addition, crop analysis indicates that wheat and rice not only have the highest volumes of water use, but also have high fractions of unsustainable use, reinforcing the urgency of more sustainable agricultural practices

In this sense, the findings of this study contribute to the understanding of the theme, demonstrating that, despite the nutritional and technological challenges already faced in the production of animal and vegetable proteins, there are still many gaps to be filled. Plant proteins emerge as a sustainable and effective alternative to animal proteins. However, nutritional limitations, such as the lower concentration of essential amino acids in plant sources, remain a challenge, especially for the large-scale production of meat analogs, due to the high costs involved. The study also highlights the potential of emerging food practices, such as composting and technological advances in agriculture, to mitigate environmental impacts. As potential future directions, we suggest the development of new processing techniques that improve the bioavailability of amino acids in plant proteins, in addition to the evaluation of the economic impact of the implementation of these practices in different regions. Additional research on the direct and indirect impacts of plant protein production, especially in relation to carbon footprint, is essential for advancing the field.

## Figures and Tables

**Figure 1 foods-13-03217-f001:**
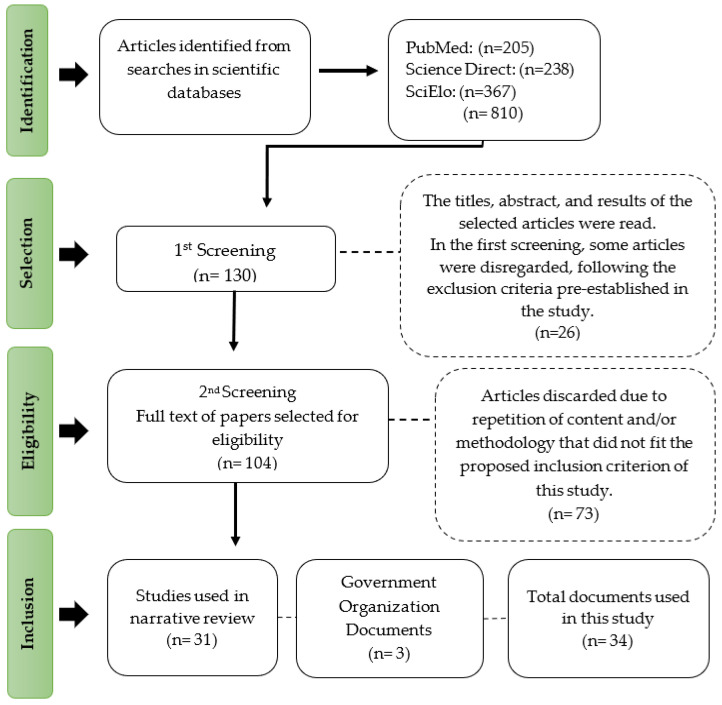
Flowchart of the scientific dataset. Source: study data.

**Table 1 foods-13-03217-t001:** Comparison among protein sources and their environmental impacts.

Protein Origin	Product	Environmental Impact	Author, Year, and Country
**Conventional meats**	Beef	7–118 kg CO_2_ eq./kg	Keoleian and Heller [44], USA
		113–166 kg CO_2_ eq./kg^−1^	Heusala [45], Finland
	Eggs	2.7 kg CO_2_ eq./kg^−1^	Fresán [46], USA
	Chicken	7.7–11.3 kg CO_2_ eq./kg^−1^	Detzel [47], Germany
	Dairy	4.38–4.95 kg CO_2_ eq./kg^−1^	Smetana [48], Germany
	Farmed fish	±60 kg CO_2_ eq./kg^−1^ ptn	
	Pork	9.1 kg CO_2_ eq./kg	Mesquita and Carvalho [49], Portugal
	Fish	10.2 kg CO_2_ eq./kg	
**Analogues to meat**	Products made from wheat	0.21 kg CO_2_ eq./100 g	Fresán [46], USA
	Soy	0.21 kg CO_2_ eq./100 g	
	Fava protein	0.23–0.58 kg CO_2_-eq./kg	Heusala [50], Finland
	Plant-based protein	2.22–35 kg CO_2_ eq./kg^−1^ptn	Detzel [47], Germany
	Legumes (soybeans, peas and lupins)	0.2–0.6 kg CO_2_ eq./kg^−1^	Heusala [50], Finland Smetana [48], Germany
	Protein potato	2.2–2.6 kg CO_2_ eq./kg^−1^	Smetana [48], Germany
	Bean	0.7–3.3 kg CO_2_ eq. /kg^−1^	
**Meat burgers**	Beef	26.6 kg CO_2_ eq./kg	Saerens [51], Germany Smetana [48], Germany Mesquita and Carvalho [49], Portugal
	Chicken	6.05 kg CO_2_ eq./kg	
	Pig	5.83 kg CO_2_ eq./kg	
	Fish	8.5 kg CO_2_ eq./kg	
**Plant-based burgers**	Pea-based from the supermarket	0.17 kg CO_2_ eq./kg	Saerens [51], Germany Smetana [48], Germany
	Made with soy from supermarket	0.19 kg CO_2_ eq./kg	
	Made from textured soy protein	0.87 kg CO_2_ eq./kg	
	Soy-based pilot product	0.06–0.1 kg CO_2_ eq./kg	
	Pumpkin seed-based pilot product	0.08–0.1 kg CO_2_ eq.	
**Edible insects**	Protein from crude edible insect biomass	3.9–29 kg CO_2_ eq./kg^−1^	Upcraft [52], United Kingdom Vauterin [53], Finland
**Cultured meat**	Cultivated meat in the global average energy scenario	14 kg CO_2_-eq./kg of meat	Sinke [54], Holland Santo [55], USA
		56 kg de CO_2_ eq./kg^−1^ protein	

Source: study data.

**Table 2 foods-13-03217-t002:** Greenhouse gas emissions from organic waste throughout the composting process.

Waste	CO_2_ Gas Emissions	References
**Cattle manure**	5–10 g dia^−1^ kg^−1^ of dry matter	Bai [65], Australia
	10.1–50.5 g/m^−2^ h^−1^	Anderson [66], USA
**Sewage sludge**	91.23–226.16 g dry/kg	Han [67], China
**Chicken manure**	29.75–51.84 g/day	Chen [68], China
**Pig manure**	208.10 g/kg of dry matter	Zeng [69], China

Source: Study data.

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
