# Peer review of "Effectiveness, Challenges, and Environmental Impacts of New Food Strategies with Plant and Animal Protein Products"

_foods, 2024, doi:10.3390/foods13203217_

Round 1

Reviewer 1 Report

Comments and Suggestions for Authors

In the submitted review manuscript (foods-3237072), the authors analysed various aspects related to the sustainability of producing protein products of plant and animal origin. They considered selected publications (scientific studies and government documents) in English, Portuguese, and Spanish from the last seven years. Some topics/problems covered were greenhouse gas emissions, climate changes, edible insects, 3D food printing, land use, water and carbon footprint, and technological innovations. Some of their advantages and/or disadvantages are also briefly discussed.

Generally speaking, this is an interesting manuscript that nicely summarises modern society's challenges in sustainable development regarding emerging problems in human nutrition.

The authors should correct the mentioned omissions.

1. The text of the manuscript is not prepared carefully enough. There are many "technical" flaws in the text: (a) not using subscripts/superscripts where it is chemically necessary (e.g., writing the carbon dioxide formula as CO2 or when marking units as kg-1); (b) 2. some parts of the text remain in Portuguese (entries in Figure 3); (c) all references are not cited completely, in the sense that the names of all authors are missing (e.g., numbers 5, 6, 25, 38, 47, 55, 57), or the style does not correspond to the instructions (e.g., 60). Please carefully study the instructions for authors and refine the whole text.

2. Line 50 should read 9 billion people, not million!

3. Parts of the text that deal with some of the analysed factors or processes are too general or do not provide "both sides of the story" (pro and contra for some proposed solution). For example, the (possible) allergenicity of edible insect proteins (lines 153161) should be mentioned (e.g., doi: 10.1016/j.molimm.2018.03.015). Or, explicitly state the advantages of the mentioned "technological innovations in agriculture" (lines 262265).

Comments on the Quality of English Language

Carefully check the quality of the (English) language in the manuscript's text.

Author Response

Dear Editor and Referee,

We would like to thank the referee for your precious time during the paper evaluation and the useful suggestions that helped us to improve the quality of the manuscript. We are grateful for all the feedback on our paper and agree with all of them.            We have added the suggestions and corrections pointed out by the referee. Questions and suggestions are in black and answers are in blue. We hope that we have now accomplished all the corrections requested by our reviewer.

 Comments to Author:

# Reviewer 1

In the submitted review manuscript (foods-3237072), the authors analysed various aspects related to the sustainability of producing protein products of plant and animal origin. They considered selected publications (scientific studies and government documents) in English, Portuguese, and Spanish from the last seven years. Some topics/problems covered were greenhouse gas emissions, climate changes, edible insects, 3D food printing, land use, water and carbon footprint, and technological innovations. Some of their advantages and/or disadvantages are also briefly discussed.

Generally speaking, this is an interesting manuscript that nicely summarises modern society's challenges in sustainable development regarding emerging problems in human nutrition.

Response: We thank our referee for the kind words and for the helpful suggestions. Everything that our referee requested, we added, and we agree that this version is much more complete with these suggestions.

The authors should correct the mentioned omissions.

  1. The text of the manuscript is not prepared carefully enough. There are many "technical" flaws in the text: (a) not using subscripts/superscripts where it is chemically necessary (e.g., writing the carbon dioxide formula as CO2 or when marking units as kg-1); (b) 2. some parts of the text remain in Portuguese (entries in Figure 3); (c) all references are not cited completely, in the sense that the names of all authors are missing (e.g., numbers 5, 6, 25, 38, 47, 55, 57), or the style does not correspond to the instructions (e.g., 60). Please carefully study the instructions for authors and refine the whole text.

Response: We are very sorry. We apologize for the inconvenience, but we believe it was a misconfiguration of our word.  We have corrected all the technical flaws to improve our paper in this new version as proposed by our referee. We also decided to remove Figure 3 avoiding any chance of Copyright.

  1. Line 50 should read 9 billion people, not million!

Response: It was corrected as the referee has suggested.

  1. Parts of the text that deal with some of the analysed factors or processes are too general or do not provide "both sides of the story" (pro and contra for some proposed solution). For example, the (possible) allergenicity of edible insect proteins (lines 153–161) should be mentioned (e.g., doi: 10.1016/j.molimm.2018.03.015). Or, explicitly state the advantages of the mentioned "technological innovations in agriculture" (lines 262–265).

Response: It was improved as our referee has suggested. We hope you approve this new version. We agree it is better than the previous sentence.

Comments on the Quality of English Language. Carefully check the quality of the (English) language in the manuscript's text.

Response: Dear referee, we have improved English.  We hope that in this 2º version it is clearer.

Reviewer 2 Report

Comments and Suggestions for Authors

Abstract

Some expressions are more complicated. It is suggested to simplify the sentence structure and make it more readable.

Introduction

Add more discussion about the existing research results of plant protein and animal protein production, so as to better show the background of the research problem.

Discussions on sustainable development goals, dietary patterns and climate change are scattered, so it is suggested to reorganize these contents to make them have a clearer logical structure.

Materials and Methods

The steps of data search and screening need to be described in more detail, especially how to determine the exclusion criteria and specific statistical analysis methods.

Discussion

Part of the discussion is simple, so more analysis and comparison should be added, especially in the environmental impact of plant protein and animal protein production, which can be supported by more literature and discussed in depth.

Suggestions such as "optimizing plant protein production" and "reducing environmental impact" are put forward, but how to implement these measures is not specified. More specific solutions or policy suggestions can be provided in this part.

Conclusion

It is necessary to further clarify the specific contributions and findings of the study, rather than just summarizing the previous contents. Suggestions or directions for future research can be put forward more clearly. In the conclusion part, the specific conclusions drawn from the research are summarized more clearly.

Figures and Tables

The figures clarity needs to be improved.

Please change the table to an editable text type instead of an uneditable picture type.

Author Response

Dear Editor and Referee,

We would like to thank the referee for your precious time during the paper evaluation and the useful suggestions that helped us to improve the quality of the manuscript. We are grateful for all the feedback on our paper and agree with all of them.            We have added the suggestions and corrections pointed out by the referee. Questions and suggestions are in black and answers are in blue. We hope that we have now accomplished all the corrections requested by our reviewer.

 Comments to Author:

# Reviewer 2

Abstract - Some expressions are more complicated. It is suggested to simplify the sentence structure and make it more readable.

Response: We are very grateful for the kindness of our reviewer. We have rewritten the abstract again to qualify it. Thank you for your suggestion.

Introduction -Add more discussion about the existing research results of plant protein and animal protein production, so as to better show the background of the research problem. Discussions on sustainable development goals, dietary patterns and climate change are scattered, so it is suggested to reorganize these contents to make them have a clearer logical structure.

 Response:  We thank our reviewer for your suggestion. We agree that it looks better the way you have recommended. It was rephrased. With your directions we understand what you meant. If something is not exactly as it was suggested, please let us know and we can write it again.

Materials and Methods - The steps of data search and screening need to be described in more detail, especially how to determine the exclusion criteria and specific statistical analysis methods.

Response: It was corrected as the referee has suggested.  We considered and followed our referee's suggestion. We hope you agree with this new version. Only, statistical analysis was not added because it was not performed, since it is not a systematic review nor meta-analysis.

Discussion - Part of the discussion is simple, so more analysis and comparison should be added, especially in the environmental impact of plant protein and animal protein production, which can be supported by more literature and discussed in depth.

Suggestions such as "optimizing plant protein production" and "reducing environmental impact" are put forward, but how to implement these measures is not specified. More specific solutions or policy suggestions can be provided in this part.

 Response: Thank you for your motivational comments and recommendations, we have worked focused on your suggestions and we hope that this version is appropriate. We adjusted the paragraphs, improved them with more discussions that will be in blue in the text. 

Conclusion -It is necessary to further clarify the specific contributions and findings of the study, rather than just summarizing the previous contents. Suggestions or directions for future research can be put forward more clearly. In the conclusion part, the specific conclusions drawn from the research are summarized more clearly.

Response:  Dear reviewer, thank you for your attentive remarks, we have worked with focus on this section of our manuscript to make it clearer. We hope that this version is suitable.

Figures and Tables -The figures clarity needs to be improved. Please change the table to an editable text type instead of an uneditable picture type.

Response: It was corrected as the referee has suggested.

Round 2

Reviewer 1 Report

Comments and Suggestions for Authors

In the revised text, the authors successfully responded to all this reviewer's remarks and suggestions, so I am pleased to propose that this version of the manuscript be accepted for publication in a special issue of Foods Journal. Congratulations!

Reviewer 2 Report

Comments and Suggestions for Authors

The manuscript has been well revised.